# Physiologically Relevant Alternative Carbon Sources Modulate Biofilm Formation, Cell Wall Architecture, and the Stress and Antifungal Resistance of *Candida glabrata*

**DOI:** 10.3390/ijms20133172

**Published:** 2019-06-28

**Authors:** Shu Yih Chew, Kok Lian Ho, Yoke Kqueen Cheah, Doblin Sandai, Alistair J.P. Brown, Leslie Thian Lung Than

**Affiliations:** 1Department of Medical Microbiology and Parasitology, Faculty of Medicine and Health Sciences, Universiti Putra Malaysia, Serdang 43400 UPM, Selangor, Malaysia; 2Department of Pathology, Faculty of Medicine and Health Sciences, Universiti Putra Malaysia, Serdang 43400 UPM, Selangor, Malaysia; 3Department of Biomedical Sciences, Faculty of Medicine and Health Sciences, Universiti Putra Malaysia, Serdang 43400 UPM, Selangor, Malaysia; 4Infectomics Cluster, Advanced Medical and Dental Institute, Universiti Sains Malaysia, Kepala Batas 13200, Pulau Pinang, Malaysia; 5MRC Centre for Medical Mycology at the University of Aberdeen, Institute of Medical Sciences, Foresterhill, Aberdeen AB25 2ZD, UK

**Keywords:** *Candida glabrata*, biofilms, cell wall, antifungal resistance, metabolic adaptation, metabolism, alternative carbon metabolism, pathogenicity

## Abstract

Flexibility in carbon metabolism is pivotal for the survival and propagation of many human fungal pathogens within host niches. Indeed, flexible carbon assimilation enhances pathogenicity and affects the immunogenicity of *Candida albicans*. Over the last decade, *Candida glabrata* has emerged as one of the most common and problematic causes of invasive candidiasis. Despite this, the links between carbon metabolism, fitness, and pathogenicity in *C. glabrata* are largely unexplored. Therefore, this study has investigated the impact of alternative carbon metabolism on the fitness and pathogenic attributes of *C. glabrata*. We confirm our previous observation that growth on carbon sources other than glucose, namely acetate, lactate, ethanol, or oleate, attenuates both the planktonic and biofilm growth of *C. glabrata*, but that biofilms are not significantly affected by growth on glycerol. We extend this by showing that *C. glabrata* cells grown on these alternative carbon sources undergo cell wall remodeling, which reduces the thickness of their β-glucan and chitin inner layer while increasing their outer mannan layer. Furthermore, alternative carbon sources modulated the oxidative stress resistance of *C. glabrata* as well as the resistance of *C. glabrata* to an antifungal drug. In short, key fitness and pathogenic attributes of *C. glabrata* are shown to be dependent on carbon source. This reaffirms the perspective that the nature of the carbon sources available within specific host niches is crucial for *C. glabrata* pathogenicity during infection.

## 1. Introduction

Nutrient assimilation is essential for the growth of all living organisms from microbes to complex multicellular organisms. Without appropriate nutritional resources, pathogens are unable to proliferate and colonise human tissues to establish infections [1]. The diverse microenvironments in the human host are both dynamic and complex in that they often contain mixtures of different carbon sources, the concentrations of which change over time [2]. In addition, the host immune system actively deprives invading pathogens of specific nutrients such as essential micronutrients and amino acids [3,4,5,6]. Successful human pathogens respond by activating robust nutrient scavenging mechanisms to ensure their survival and propagation within the host [7,8,9,10].

*Candida* species are amongst the most common cause of hospital-acquired systemic mycoses. The spectrum of disease caused by *Candida* species ranges from superficial infections like oral thrush and vulvovaginal candidiasis (VVC) to lethal invasive infections like invasive candidiasis. *Candida albicans* is the predominant causative agent of life-threatening systemic candidiasis. Yet there is an alarming increase in the incidence of candidiasis caused by *Candida glabrata* and other non-*C. albicans Candida* (NCAC) species [11] due in part to the intrinsic resistance of these species to azole antifungal drugs [12,13].

*Candida* cells colonising different anatomical sites encounter microenvironments with different carbon source availabilities. Glucose might be an efficient carbon and energy source for the growth of microorganisms in vitro and be present in the bloodstream, but it is scarce in host niches such as the lower gastrointestinal tract or inside macrophages. Indeed, *Candida* species such as *C. albicans* and *C. glabrata* display nutrient starvation responses following phagocytosis by macrophages [3,7,9,14]. For this reason, efficient metabolic adaptation is integral to the pathogenicity of *Candida* species as well as classical virulence factors such as cellular morphogenesis, phenotypic switching, secreted aspartyl proteinases, adhesins, invasins, and biofilm formation [15,16]. Genes encoding enzymes involved in gluconeogenesis, the glyoxylate cycle and fatty acid β-oxidation, are induced following phagocytosis, suggesting that the microenvironment inside macrophages is deficient in glucose and that the engulfed *Candida* cell must switch to alternative carbon sources. Also, the colon is deficient in glucose, meaning that lactate assimilation is essential for the propagation of *C. glabrata* cells in the gastrointestinal tract [9]. *C. glabrata* is also able to metabolise acetic acid produced by lactic acid bacteria in the vagina [17]. 

Adaptation to different carbon sources affects other virulence-related properties. For example, *C. albicans* cells that metabolise lactate have a thinner cell wall with a reduced β-glucan and chitin inner layer when compared with glucose-grown control cells [2]. These changes in cell wall architecture in lactate-grown cells enhances the resistance of *C. albicans* to multiple antifungal drugs as well as osmotic and cell wall stresses. Furthermore, *C. albicans* cells exposed to lactate are less visible to innate immune cells than control untreated cells [18,19]. Moreover, lactate-grown cells are more efficient in killing the host’s macrophages. In contrast, *C. glabrata* cells that assimilate acetate are more susceptible to fluconazole, albeit in the presence of glucose, generate less robust biofilms, and are more susceptible to macrophage killing [20].

Despite the recent increase in the incidence of candidiasis caused by *C. glabrata* [11,21], relatively little is known about the physiological behaviour of *C. glabrata* in the host microenvironments. Therefore, in this study we have compared the effects of glucose and alternative carbon utilisation on the growth, biofilm formation, cell wall architecture, stress response, and antifungal susceptibility of *C. glabrata*.

## 2. Results

### 2.1. Alternative Carbon Sources Affect the Planktonic and Biofilm Growth of C. glabrata

As a starting point, the planktonic growth of *C. glabrata* was compared on glucose and alternative carbon sources using a plate-based microtiter growth assay. *C. glabrata* grew on all the carbon sources tested in this study, including glucose, acetate, lactate, ethanol, glycerol, and oleic acid (Figure 1a), thereby recapitulating our previous observations [22]. Predictably, the planktonic behaviour of all cultures was similar during the first 4 h of incubation. On glucose, *C. glabrata* entered exponential phase after 4 h whereas, on alternative carbon sources, the *C. glabrata* grown remained in the lag phase for longer (over 6 h). Growth was strong on glucose (doubling time, T_d_ ~ 2 h) and glycerol (T_d_ ~ 6 h), medium on lactate (T_d_ ~ 8 h), and less strong on oleic acid (T_d_ ~ 10 h), ethanol (T_d_ ~ 10 h) and acetate (T_d_ ~ 14 h). Although *C. glabrata* utilises a variety of alternative carbon sources, we infer it has a preference for glycerol and lactate over the other alternative carbon sources tested.

Next, we compared the biofilm growth of *C. glabrata* on the same carbon sources (Figure 1b). With the exception of glycerol, biofilm growth was reduced significantly on the alternative carbon sources tested when compared with the glucose control. Substantial reductions in biofilm growth were observed for *C. glabrata* on acetate (52.4%), lactate (30.3%), ethanol (26.9%), and oleic acid (27.3%), thereby confirming our previous report [23]. The ultrastructure of these biofilms was examined by SEM (Figure 2). The *C. glabrata* biofilms comprised only blastoconidia: no filamentous cells were observed. The aggregates of *C. glabrata* cells were smaller in biofilms grown in alternative carbon sources, consistent with the reduced growth on these carbon sources. 

### 2.2. Alternative Carbon Sources Influence C. glabrata Cell Wall Architecture

The effect of alternative carbon sources on the cell wall architecture of *C. glabrata* was investigated by TEM. This revealed significant differences in cell wall architecture and thickness depending on the carbon source (Figure 3a). The thickness of the inner cell wall, which comprises largely β-glucan and chitin, was reduced significantly following growth on acetate (57.42 ± 10.55 nm), lactate (51.09 ± 11.80 nm), ethanol (57.94 ± 9.48 nm), and oleic acid (56.72 ± 7.73 nm) when compared with the glucose control (81.70 ± 6.68 nm) (Figure 3b). The effect on the inner cell wall was less dramatic for glycerol (71.73 ± 12.40 nm). In contrast, the thickness of the outer mannan layer of the cell wall increased significantly when *C. glabrata* cells were grown on the alternative carbon sources: acetate (24.58 ± 3.80 nm), lactate (31.04 ± 7.17 nm), glycerol (25.44 ± 2.82 nm) and oleic acid (24.61 ± 2.78 nm), and glucose control (20.10 ± 3.41 nm) (Figure 3c). The increase for ethanol was marginal (22.93 ± 2.24 nm). In short, utilisation of some alternative carbon sources led to changes in the cell wall architecture of *C. glabrata,* reducing the thickness of β-glucan and chitin inner layer whilst increasing the thickness of the outer mannan layer.

### 2.3. Alternative Carbon Sources Reduce the Susceptibility of C. glabrata to Antifungal Drugs 

To investigate the impact of carbon sources on the susceptibility of *C. glabrata* to antifungal drugs, the capacity of *C. glabrata* cells grown on glucose or alternative carbon sources to withstand amphotericin B was examined using a broth microdilution assay. All glucose-grown *C. glabrata* cells were killed by treatment with 0.5 μg/mL amphotericin B (Figure 4). Significantly, growth on alternative carbon sources reduced the susceptibility of *C. glabrata* cells towards amphotericin B, resisting its fungicidal activity to varying degrees (Figure 4). Lactate- and oleate-grown *C. glabrata* cells were most resistant to amphotericin B whilst acetate-, ethanol- and glycerol-grown cells were only slightly more resistant than the glucose-grown control cells. Our observations suggest that *C. glabrata* cells growing on alternative carbon sources in vivo may be less sensitive to antifungal therapy.

### 2.4. Alternative Carbon Sources Modulate the Oxidative Stress Resistance of C. glabrata

Finally, we investigated the impact of alternative carbon sources on the oxidative stress resistance of *C. glabrata* by examining the sensitivity of this pathogen to hydrogen peroxide (H_2_O_2_). As shown in Figure 5, glucose-grown *C. glabrata* is relatively resistant to high level of H_2_O_2_, which is in agreement with previous findings [23]. However, when *C. glabrata* was grown on alternative carbon sources, it was much more susceptible to H_2_O_2_ treatment (Figure 5). We conclude that carbon source impacts significantly upon the oxidative stress resistance of *C. glabrata*.

## 3. Discussion

Metabolic adaptation is thought to affect the pathogenicity of *C. albicans* at multiple levels, influencing the expression of virulence factors as well as fitness attributes and also affecting immune surveillance [15]. In *C. glabrata*, the assimilation of alternative carbon sources is known to be essential for proliferation in glucose-poor niches [14,22,24]. However, the impact of alternative carbon sources upon *C. glabrata* fitness attributes and virulence phenotypes had not been addressed. Therefore, we examined the effects upon biofilm formation, cell wall architecture, and antifungal susceptibility in this study. 

First we confirmed that glucose is the most efficient carbon source for the planktonic growth of *C. glabrata* of those we tested [22]. Glucose feeds directly into glycolysis, generating intermediates for growth plus energy through a combination of fermentation, the tricarboxylic acid (TCA) cycle, and respiration [25,26]. In contrast, growth on glycerol or lactate requires functional gluconeogenesis to generate the hexoses and pentoses required for cell wall and nucleic acid biosynthesis [27]. The assimilation of ethanol or acetate also requires a combination of the glyoxylate and TCA cycles to generate anabolic precursors and energy in addition to gluconeogenesis. Moreover, the utilisation of oleate also requires fatty acid β-oxidation. In contrast to *C. albicans* [16,28], *C. glabrata* expresses most of the requisite metabolic enzymes for alternative carbon assimilation only when they are needed [27]. Hence, the diauxic shift from the utilisation of glucose to an alternative carbon source requires time for adaptation, as demonstrated by the longer lag phase during the initiation of *C. glabrata* growth on alternative carbon sources (Figure 1a and Figure 2).

Next, we examined biofilm formation, revealing that *C. glabrata* forms weaker biofilms on acetate, lactate, ethanol, and oleate when compared with the glucose-grown control (Figure 1b). This observation is in concordance with previous reports showing that *C. albicans* biofilms are greatly reduced during growth on lactate [29] and that *C. glabrata* biofilms are diminished by the addition of acetate to glucose-containing medium [20]. Interestingly, glycerol does not seem to affect the growth of *C. glabrata* in biofilms. This might relate to the essential regulatory role of glycerol in the expression of biofilm-related functions such as adhesins [30]. Glycerol is required for the synthesis of the glycosylphosphatidylinositol (GPI) anchor, which links adhesins and other mannoproteins to the cell wall and potentially assists in biofilm formation [31]. 

Several factors, including carbon source, hypoxia, pH, and micronutrient limitation, induce cell wall remodelling in *C. albicans* [2,32,33,34]. We show that changes in carbon source also induce changes in cell wall architecture of *C. glabrata* (Figure 3). Our data indicate that, with the possible exception of glycerol, growth on carbon sources other than glucose is associated with reductions in the thickness of the inner cell wall and, with the possible exception of ethanol, increases in the thickness of outer mannan layer of the cell wall. This differs slightly from the effects of carbon source on the *C. albicans* cell wall. According to Ene et al. (2012), both the inner and outer layers of the cell wall are thinner for lactate-grown *C. albicans* cells when compared with their glucose-grown controls [2]. The mechanisms by which carbon sources affect cell wall architecture remain unclear. Nevertheless, these are likely to include direct effects upon the regulation of cell wall biosynthetic genes combined with different rates of provision of essential cell wall precursors via glycolytic versus gluconeogenic metabolism [2]. This could account for the formation of thinner inner cell walls during growth on alternative carbon sources. The extended mannan fibrils in the outer cell wall could conceivably be due to changes in the balance between the extension versus branching of mannan sidechains; environmental signals have been suggested to modulate structure of mannan fibrils [2,35,36]. The activation of the cell wall integrity pathway is a common response to environmental challenges in human fungal pathogens [37,38,39,40,41]. Whatever the underlying mechanisms, based on recent findings in *C. albicans* [18,19,34], the changes in the *C. glabrata* cell wall that accompany carbon source adaptation are likely to affect the visibility of this pathogen to our innate immune defences.

Changes in carbon source are known to affect the resistance of *C. albicans* to environmental stresses [2,42] and to affect stress signalling mechanisms in this pathogen [43]. *C. glabrata* displays high levels of intrinsic resistance to such stresses [23], which is related to its evolution as a pathogen that can thrive within the challenging microenvironment of the macrophage [44]. Phylogenetically, *C. glabrata* is more closely related to *S. cerevisiae* than *C. albicans* [45]. In *S. cerevisiae*, the presence of glucose represses the core stress response via protein kinase A-mediated phosphorylation of the transcription factors Msn2 and Msn4 [46,47]. *Candida glabrata* has been reported to display a core stress response that resembles that of *S. cerevisiae* [48] and, on this basis, might be expected to be more sensitive to oxidative stress during growth on glucose. However, our data (Figure 5) suggest that *C. glabrata* behaves more like *C. albicans,* which displays glucose-enhanced oxidative stress resistance [49]. This would be consistent with the possibility that, like *C. albicans*, *C. glabrata* has evolved an anticipatory response whereby exposure to glucose helps to protect this pathogen against the impending oxidative stress associated with phagocytic assaults [49,50]. The effects of carbon source upon the susceptibility of *C. glabrata* to amphotericin B (Figure 4) are less surprising and are significant in terms of antifungal therapy. Therefore, we argue that carbon source adaptation in *C. glabrata* is likely to affect the efficacy of clinical treatments as well as fungal pathogenicity.

## 4. Materials and Methods

### 4.1. Strain and Growth Condition

The reference strain *C. glabrata* ATCC 2001 was used in this study (American Type Culture Collection, Manassas, VA, USA). Standard culture media were used, including YPD (Becton, Dickinson and Company, Franklin Lakes, NJ, USA): yeast extract (1%, *w*/*v*), peptone (2%, *w*/*v*), glucose (2%, *w*/*v*), agar (1.5%, *w*/*v*), and yeast nitrogen base (YNB) without amino acids (Becton, Dickinson and Company, USA): yeast nitrogen base (0.67%, *w*/*v*), ammonium sulfate (0.5%, *w*/*v*). Synthetic complete (SC) media were prepared with YNB without amino acids, supplemented with complete supplement mixture (0.2%, *w*/*v*) (Formedium, Hunstanton, UK), glucose (2%, *w*/*v*), and agar (2%, *w*/*v*). In addition, glucose was replaced with alternative carbon sources: acetate (2%, *w*/*v*), lactate (2%, *v*/*v*), ethanol (2%, *v*/*v*), glycerol (2%, *v*/*v*), or oleic acid (0.2%, *w*/*v*) (Sigma-Aldrich, St. Louis, MO, USA) as the sole carbon source in SC media [2,51].

### 4.2. Planktonic Growth Assay

Growth of *C. glabrata* in glucose and alternative carbon sources was assessed by plate-based microtiter growth assay [22]. Briefly, *C. glabrata* was grown in YPD overnight at 37 °C, harvested and washed twice with phosphate-buffered saline (PBS), pH 7.4 before resuspended into fresh SC media supplemented with glucose, acetate, lactate, ethanol, glycerol, and oleic acid as the sole carbon source (OD_600nm_ of 0.1). A volume of 200 µL of *C. glabrata* cell suspension was transferred into a sterile 96-well plate and incubated at 37 °C. Growth of *C. glabrata* was monitored for 72 h by measuring OD_600nm_ with a microtiter plate reader (Dynex Technologies, Chantilly, VA, USA).

### 4.3. Biofilm Formation Assay

Biofilm formation of *C. glabrata* in glucose and alternative carbon sources was determined by using 2,3-bis-(2-methoxy-4-nitro-5-sulfophenyl)-2*H*-tetrazolium-5-carboxanilide (XTT) reduction assay as previously described [52]. Overnight culture of *C. glabrata* was harvested, washed twice with PBS, pH 7.4, and resuspended in SC media supplemented with glucose, acetate, lactate, ethanol, glycerol, and oleic acid as sole carbon source (OD_600nm_ of 0.1). A volume of 100 µL of *C. glabrata* cell suspension was dispensed into selected wells of a pre-sterilised, clear and flat bottom 96-well plate. The 96-well plate was covered with its original lid, sealed with parafilm, and incubated for 48 h at 37 °C. Following 48 h of incubation, the plate was washed twice with PBS, pH 7.4 to remove the planktonic cells. A volume of 100 µL of XTT/menadione solution (0.5 g/L XTT, 10 mM menadione, Sigma-Aldrich, USA) was added to the formed biofilm and incubated in the dark for 3 h at 37 °C. After 3 h of incubation, 80 µL of the solution was transferred to a new 96-well plate and biofilm formation of *C. glabrata* was quantified by measuring OD_490nm_ of the biofilm using a microtiter plate reader.

### 4.4. Visualisation of Biofilm Structures

Biofilm structures of *C. glabrata* grown in glucose and alternative carbon sources were visualised using SEM. Briefly, overnight culture of *C. glabrata* was centrifuged, washed twice with PBS, pH 7.4, and resuspended into fresh SC media supplemented with glucose, acetate, lactate, ethanol, glycerol, and oleic acid (OD_600nm_ of 0.1). Cell suspension was dispensed into a pre-sterilised, clear and flat bottom 6-well polystyrene microtiter plate (Becton, Dickinson and Company, USA) with Nunc Thermanox coverslips (Thermo Fisher Scientific, Waltham, MA, USA). The plate was covered with its original lid, sealed with parafilm, and incubated for 48 h at 37 °C. After 48 h of incubation, coverslips were fixed in 4% (*v*/*v*) glutaraldehyde (Agar Scientific, Stansted, UK) for 6 h at 4 °C, followed by washing with 0.1 M sodium cacodylate buffer, pH 7.2 (Agar Scientific, UK) for three times of 10 min each. Coverslip was post-fixed in 1% osmium tetroxide (*w*/*v*) (Agar Scientific, UK) for 2 h at 4 °C, washed with 0.1 M sodium cacodylate buffer for three times of 10 min each, and dehydrated with increasing concentration of acetone (Friendemann Schmidt, Parkwood, Australia): 35% (*v*/*v*) for 10 min, 50% (*v*/*v*) for 10 min, 75% (*v*/*v*) for 10 min, 95% (*v*/*v*) for 10 min and 100% (*v*/*v*) for 15 min (three times). Prior to viewing, coverslips were air-dried in EM CPD030 critical point dryer (Leica, Wetzlar, Germany), mounted onto aluminium stubs and sputter coated with gold using EM SCD050 sputter coater (Leica, Germany). Biofilm structures of *C. glabrata* grown in glucose and alternative carbon sources were observed with LEO 1455 VP SEM (Carl-Zeiss, Oberkochen, Germany).

### 4.5. Visualisation of Cell Wall Architecture

Cell wall architectures of *C. glabrata* grown in glucose and alternative carbon sources were investigated by TEM. Briefly, overnight culture of *C. glabrata* was centrifuged, washed twice with PBS, pH 7.4, and resuspend in SC media supplemented with glucose, acetate, lactate, ethanol, glycerol, and oleic acid (OD_600nm_ of 0.1). After 48 h of incubation at 37 °C, cells were fixed in 4% (*v*/*v*) glutaraldehyde for 6 h at 4 °C. Subsequently, fixative was removed, and horse serum was added to coagulate the cell pellets. After overnight incubation, the coagulated cell pellets were diced into 1 mm^3^ pieces and fixed in 4% (*v*/*v*) glutaraldehyde for another 2 h at 4 °C. Following three times of washing with 0.1 M sodium cacodylate buffer, pH 7.2, the samples were post-fixed in 1% osmium tetroxide (*w*/*v*) for 2 h at 4 °C, washed three times with 0.1 M sodium cacodylate buffer, and dehydrated with increasing concentration of acetone: 35% (*v*/*v*) for 10 min, 50% (*v*/*v*) for 10 min, 75% (*v*/*v*) for 10 min, 95% (*v*/*v*) for 10 min and three times of 100% (*v*/*v*) for 15 min. The samples were infiltrated with resin: acetone (1:1) mixture (Agar Scientific, UK) for 1 h, followed by resin: acetone (3:1) mixture for 2 h, 100% resin for overnight and 100% fresh resin for 2 h. Finally, the samples were embedded in BEEM capsules (Agar Scientific, UK) with 100% fresh resin and left to polymerisation for 48 h at 60 °C. Ultrathin sections were cut using EM UC7 ultramicrotome (Leica, Germany) at a thickness of 80 nm. Samples were visualised and imaged with a JEM-2100F field emission electron microscope (JEOL, Tokyo, Japan). Thickness of cell wall layers of *C. glabrata* (chitin plus β-glucan and mannan) were quantified using Image J by averaging 20 measurements of each cell grown in glucose and alternative carbon sources (n = 10 cells).

### 4.6. Antifungal Susceptibility Assay

Minimum inhibitory concentration (MIC) of amphotericin B against *C. glabrata* ATCC 2001 was determined by broth microdilution method according to the Clinical and Laboratory Standards Institute (CLSI) M27-A3 document with slight modification by replacing Roswell Park Memorial Institute (RPMI) 1640 medium with SC medium with 2% glucose. The MIC was confirmed to be 0.5 μg/mL. Antifungal susceptibility assay was performed according to the procedure as previously described [53]. Briefly, overnight culture of *C. glabrata* was resuspended in SC media supplemented with glucose and alternative carbon sources to OD_600nm_ of 0.1 and regrown to OD_600nm_ of 0.5. Cells growing on acetate, lactate, ethanol, and oleic acid were concentrated by centrifugation to achieve this OD. The cell suspensions were treated with 0.5 μg/mL of amphotericin B for 24 h at 37 °C. Colony-forming-unit (CFU) was determined after incubation and the viability of *C. glabrata* was defined as (CFU of amphotericin B-treated sample / CFU of untreated sample) × 100%.

### 4.7. Oxidative Stress Assay

To assess the impact of carbon source on oxidative stress resistance, overnight cultures of *C. glabrata* grown in YPD were harvested and the cells washed twice with phosphate-buffered saline (PBS), pH 7.4, before resuspension in fresh SC media supplemented with glucose, acetate, lactate, ethanol, glycerol, or oleic acid as sole carbon source (OD_600nm_ of 0.1). Then 200 µL of each *C. glabrata* cell suspension were transferred into a 96-well microtiter plate and H_2_O_2_ (Sigma-Aldrich, USA) added to a final concentration of 0 or 10 mM. Growth was then monitored for 24 h by measuring OD_600nm_ with a microtiter plate reader (Thermo Fisher Scientific, USA).

### 4.8. Statistical Analyses

Statistical analyses were performed using GraphPad Prism Version 7.0 Software (GraphPad Software Inc., San Diego, CA, USA). All experiments were performed at least in three replicates and all data were expressed as mean values from all replicates with the corresponding standard deviations (SD). Differences between control and samples were assessed by unpaired t-test and a *p* < 0.05 was considered to be statistically significant. All significant differences were indicated in the figures, with *, **, and *** indicating *p* < 0.05, *p* < 0.01 and *p* < 0.001, respectively.

## 5. Conclusions

Our data suggest the importance of carbon source adaptation in modulating important fitness and virulence attributes of *C. glabrata*. These include changes in stress resistance and the cell wall. These observations imply that, as *C. glabrata* cells adapt to the nutrients available within their local microenvironment within the host, this affects the pathogenicity of this fungus and potentially immune responses against the fungus. Furthermore, our data indicate that carbon source adaptation affects susceptibility of this pathogen to antifungal therapy. 

## Figures and Tables

**Figure 1 ijms-20-03172-f001:**
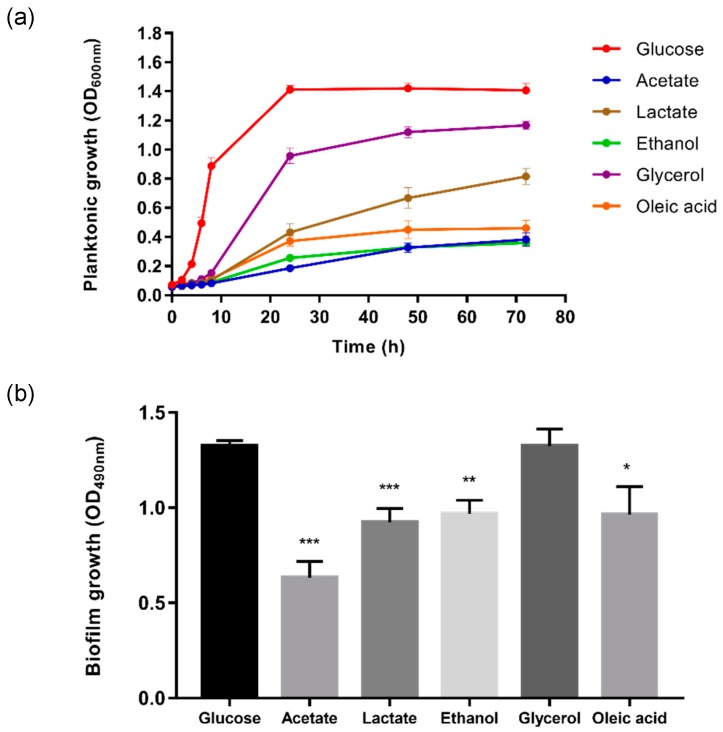
Alternative carbon sources modulate the planktonic and biofilm growth of *C. glabrata*. (**a**) Planktonic growth of *C. glabrata* in synthetic complete (SC) media supplemented with 2% glucose, 2% acetate, 2% lactate, 2% ethanol, 2% glycerol, or 0.2% oleic acid and incubated for 72 h at 37 °C; (**b**) Biofilm formation of *C. glabrata* in SC media supplemented with 2% glucose, 2% acetate, 2% lactate, 2% ethanol, 2% glycerol, or 0.2% oleic acid. Results were presented as means ± SD. * *p* < 0.05, ** *p* < 0.01 and *** *p* < 0.001 were considered statistically significant relative to the control (2% glucose). All experiments were conducted in triplicate, and each independent experiment was repeated three times.

**Figure 2 ijms-20-03172-f002:**
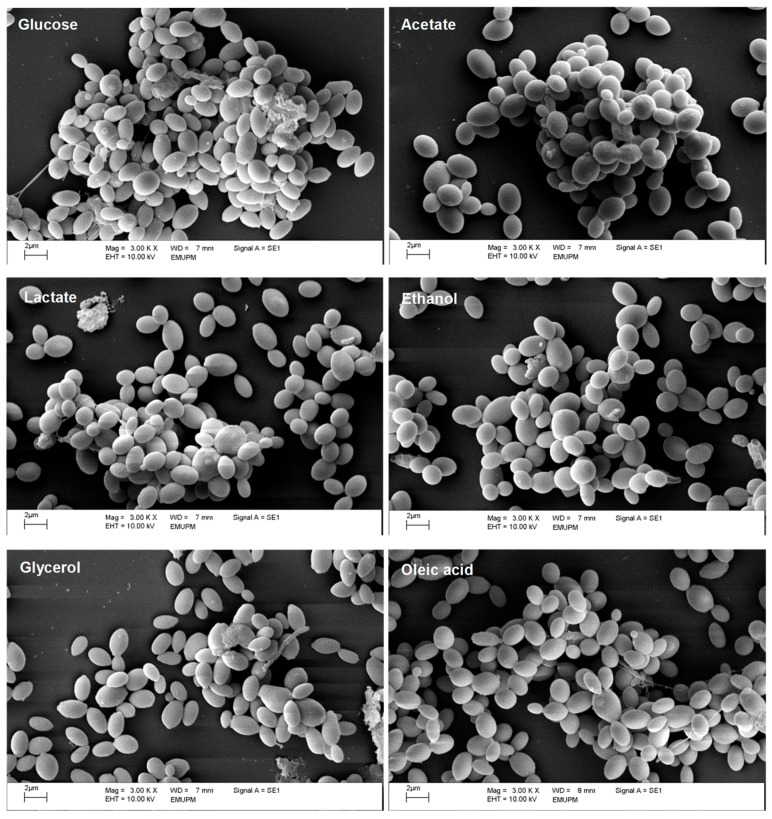
Effect of glucose and alternative carbon sources on the biofilm structures of *C. glabrat.* Representative scanning electron photomicrographs from 48 h biofilms of *C. glabrata* grown on glucose and alternative carbon sources, viewed under scanning electron microscopy (SEM) at a magnification of 3000×. Scale bar represents 2 µm.

**Figure 3 ijms-20-03172-f003:**
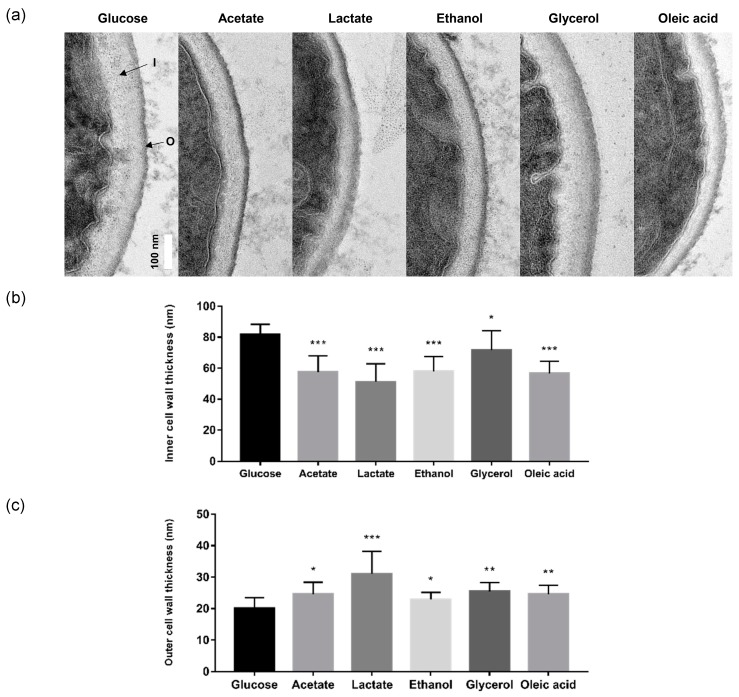
Alternative carbon sources alter the ultrastructure of the *C. glabrata* cell wall. (**a**) Representative transmission electron microscopy (TEM) images of cell wall architectures from *C. glabrata* grown on glucose and alternative carbon sources, [I] indicates inner cell wall layer (β-glucan and chitin) and [O] indicates outer cell wall layer (mannan). Scale bar represents 100 nm. (**b**) Quantification of thickness of the inner cell wall. (**c**) Quantification of the thickness of the outer cell wall. Results are presented as means ± SD from 10 individual cells by averaging 20 measurements obtained from the cell periphery of each *C. glabrata* cells. * *p* < 0.05, ** *p* < 0.01 and *** *p* < 0.001 were considered statistically significant relative to glucose-grown cells.

**Figure 4 ijms-20-03172-f004:**
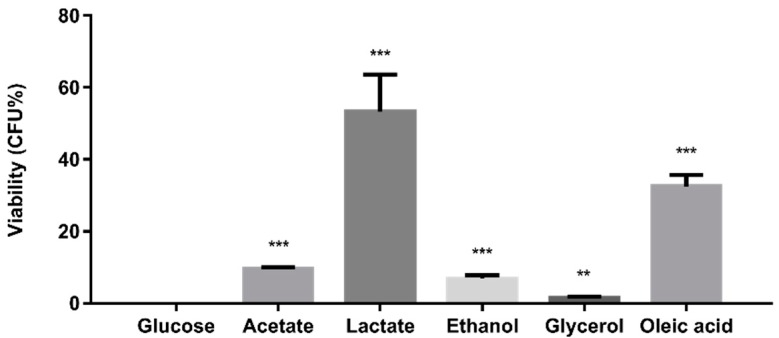
Alternative carbon sources reduce the susceptibility of *C. glabrata* to amphotericin B. Results are presented as means ± SD. ** *p* < 0.01 and *** *p* < 0.001 were considered statistically significant relative to glucose-grown cells.

**Figure 5 ijms-20-03172-f005:**
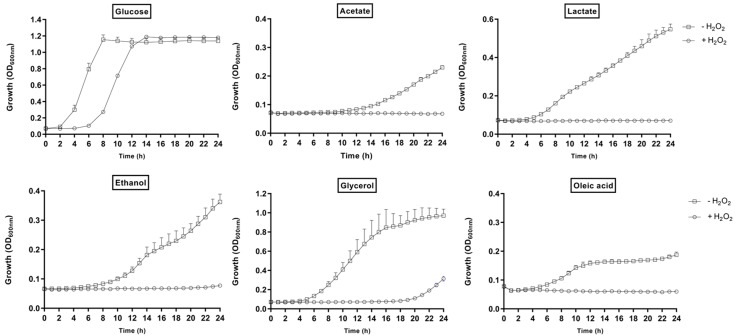
Alternative carbon sources affect the oxidative stress resistance of *C. glabrata*. Growth of *C. glabrata* in SC media supplemented with glucose or alternative carbon sources in the presence or absence of H_2_O_2_ was monitored for 24 h at 37 °C.

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
