# Peer review of "Physiologically Relevant Alternative Carbon Sources Modulate Biofilm Formation, Cell Wall Architecture, and the Stress and Antifungal Resistance of Candida glabrata"

_ijms, 2019, doi:10.3390/ijms20133172_

Round 1
Reviewer 1 Report
Overall, the manuscript is well written, and the experiments are logically presented. The results are an important step towards unravelling the importance of carbon metabolism in the pathogenicity of Candida glabrata. Thus, I support the publication of this manuscript in IJMS upon addressing the comments below.
Figure 1A:
Initially, I thought that that the final biomass differences at 72 h may have been due to carbon limitation in the acetate and ethanol samples at later time points, as the same % v/v was used across all carbon sources (except oleic acid). However, the final concentration of these carbon sources – glucose (110 mM), acetate (250 mM) and ethanol (450 mM) seem to be in excess of what is required to reach a high final OD600.
Considering the high concentrations of these carbon sources present, and the fact that the base media used is identical for all carbon sources, it is unlikely that carbon has become a limiting factor by 72 h. Therefore, what limitation do you believe accounts for the lower final biomass (OD600) of these cells at 72 h compared to glucose/glycerol? Can this final OD600 defect be overcome by spiking glucose or glycerol into the low-final yield samples?
Is it possible that the lower final OD from planktonic/biofilm growth in acetate, ethanol and oleic acid are due to differences in the growth rate? Do these cells eventually reach a comparable OD600 to glucose/glycerol if the incubation time is extended beyond 72 h?
The symbols used to differentiate Glucose and Glycerol are very similar on Figure 1 (A).
Lines 177-180: “In contrast, the assimilation of alternative carbon sources from ethanol, acetate, pyruvate, lactate or glycerol requires a combination of the glyoxylate and TCA cycles to generate anabolic precursors and energy, as well as gluconeogenesis to generate hexoses and pentoses required for cell wall and nucleic acid biosynthesis”.
I think it would be helpful to add a sentence here about the glyoxylate cycle not being required for glycerol assimilation in C. glabrata, unlike in C. albicans.
Lines 307-308 and 314-316: ‘Briefly, overnight culture of C. glabrata was resuspended in SC media supplemented with glucose and alternative carbon sources to OD600nm of 0.1 and regrown to OD600nm of 0.5.’
According to Fig 1A, cells grown on acetate only reach an OD600 of 0.2-0.3 during growth on acetate, ethanol or oleic Acid? Were these cells concentrated by centrifugation?
Author Response
RESPONSES TO REVIEWER 1 COMMENTS
Dear Editor,
We appreciate the valuable and constructive comments provided by the REVIEWERS. We have taken them fully into account in our revised manuscript (changes to the text highlighted in yellow). Our responses to the Reviewer’s reports are as follows:
Point 1:
Figure 1A:
Initially, I thought that that the final biomass differences at 72 h may have been due to carbon limitation in the acetate and ethanol samples at later time points, as the same % v/v was used across all carbon sources (except oleic acid). However, the final concentration of these carbon sources – glucose (110 mM), acetate (250 mM) and ethanol (450 mM) seem to be in excess of what is required to reach a high final OD600.
Considering the high concentrations of these carbon sources present, and the fact that the base media used is identical for all carbon sources, it is unlikely that carbon has become a limiting factor by 72 h. Therefore, what limitation do you believe accounts for the lower final biomass (OD600) of these cells at 72 h compared to glucose/glycerol? Can this final OD600 defect be overcome by spiking glucose or glycerol into the low-final yield samples?
Is it possible that the lower final OD from planktonic/ biofilm growth in acetate, ethanol and oleic acid are due to differences in the growth rate? Do these cells eventually reach a comparable OD600 to glucose/ glycerol if the incubation time is extended beyond 72 h?
Response 1:
The Reviewer makes a good point: the OD600 at 72 h is not the best measure of differences in growth, because some cultures may not have reached their “final biomass” (OD600) at this point. Therefore, we have replaced this metric with doubling times for the different carbon sources, and added information about the lag times as well (Lines 87 - 92).
Point 2:
The symbols used to differentiate glucose and glycerol are very similar on Figure 1 (A).
Response 2:
Figure 1 has been amended to address this concern (Line 95).
Point 3:
Lines 177-180: “In contrast, the assimilation of alternative carbon sources from ethanol, acetate, pyruvate, lactate or glycerol requires a combination of the glyoxylate and TCA cycles to generate anabolic precursors and energy, as well as gluconeogenesis to generate hexoses and pentoses required for cell wall and nucleic acid biosynthesis”.
I think it would be helpful to add a sentence here about the glyoxylate cycle not being required for glycerol assimilation in C. glabrata, unlike in C. albicans.
Response 3:
This point has now been clarified (Lines 179-183), as requested.
Point 4:
Lines 307-308 and 314-316: ‘Briefly, overnight culture of C. glabrata was resuspended in SC media supplemented with glucose and alternative carbon sources to OD600nm of 0.1 and regrown to OD600nm of 0.5.’
According to Fig 1(A), cells grown on acetate only reach an OD600 of 0.2 - 0.3 during growth on acetate, ethanol or oleic Acid? Were these cells concentrated by centrifugation?
Response 4:
Yes, the cells were grown in a larger volume and concentrated by centrifugation. This is now clarified in the text (Lines 311 - 313).

Reviewer 2 Report
In this study the authors study how the different carbon sources modulate the fitness and pathogenic attributes of Candida glabrata. The ultimate goal is to propose that the different carbon sources in the specific host niches is crucial for pathogenicity during infection. The authors demonstrated that biofilm formation, cell wall architecture and antifungal sensitivity are altered in the different carbon sources. They would like to infer that these changes affect the pathogenicity. However, no pathogenic assay was shown/done that will really demonstrate the importance of the carbon source.
Some results are interesting, like the sensitivity to antifungal drugs, although some additional control and/or experiments will be required before publication.
1. In Figure 1, the authors shown the growing curves in different carbon sources. The choice of symbols and line is not really adequate, and it is very difficult to read the graph. Some dashes lines, white symbols and other tricks will help to follow the results.
2. Page 3, lines 87-90. The sentence “After the lag phase of about 4h…” is no correct, since the lag phase is 4h only in glucose media. In the other media the lag phase is longer.
The OD numbers for each carbon sources described in the text is not really clear when they were taken. It is important to compare OD in exponentially growing cells and not in lag or stationary phase.
3. In Figure 3 the authors shown the EM images of the cell wall in different carbon sources. The inner membrane thickness is reduced in the different carbon sources (around 30-37% in most of them). However, in glycerol while the difference is statistically significant the reduction is around 12% and the biological relevance is disputable. The text should be fixed accordingly.
4. The outer cell wall thickness increased in several alternative carbon sources, however in the case of the ethanol again the difference is about 12% and might not be biologically relevant.
5. The EM images to determine the cell wall thickness is a good experiment, but other assay could help to further prove this observation. Different assays (such as biochemical fractionation or staining of b-glucans/chitin and mannose) can be used to further confirm those results.
6. The most interesting result is the resistance to amphotericin B in the alternative carbon sources. Repeat the experiment with another antifungal drug will be interesting to see whether this is a general resistance feature.
7. Figure 5, the authors assay the resistance of C. glabrata to H2O2. Here the authors grow the cells for a few hours in the alternative carbon source and then plate them in YPD+H2O2. This kind of experiment are frequently performed in a different way. The cells quickly adapt to the glucose in the YPD plates. Therefore, they should plate the cells in YP+alternative carbon source+H2O2 and compare to the YPD and YPD+H202 plates. Nevertheless, this will imply to compare cells in different plates and still will not be ideal. Therefore, growing curves (as in Fig.1) in presence of alternative carbon sources and with or without H2O2 might be more appropriate in this case.
8. The authors do not perform pathogenic assays after growing cells in alternative carbon sources. In some species, infections using vegetables or fruits are commonly use in the laboratory. I am not sure how easy is to perform this assay with C. glabrata, but will be pretty interesting to perform.
Author Response
RESPONSES TO REVIEWER 2 COMMENTS
Dear Editor,
We appreciate the valuable and constructive comments provided by the REVIEWERS. We have taken them fully into account in our revised manuscript (changes to the text highlighted in yellow). Our responses to the Reviewer’s reports are as follows:
Point 1:
In Figure 1, the authors shown the growing curves in different carbon sources. The choice of symbols and line is not really adequate, and it is very difficult to read the graph. Some dashes lines, white symbols and other tricks will help to follow the results.
Response 1:
Figure 1 has been amended to address this concern (Line 95).
Point 2:
Page 3, lines 87-90. The sentence “After the lag phase of about 4h…” is no correct, since the lag phase is 4h only in glucose media. In the other media the lag phase is longer.
The OD numbers for each carbon sources described in the text is not really clear when they were taken. It is important to compare OD in exponentially growing cells and not in lag or stationary phase.
Response 2:
The point has now been addressed (Lines 87-92). The Reviewer makes a good point: the OD600 at 72 h is not the best measure of differences in growth. Therefore, we have replaced this metric with doubling times in exponentially growing cells for the different carbon sources, and added information about the lag times as well.
Point 3:
In Figure 3 the authors shown the EM images of the cell wall in different carbon sources. The inner membrane thickness is reduced in the different carbon sources (around 30-37% in most of them). However, in glycerol while the difference is statistically significant the reduction is around 12% and the biological relevance is disputable. The text should be fixed accordingly.
Response 3:
The Reviewer is correct to point out that the reduction in inner cell wall thickness on glycerol, whilst statistically significant, is small. We address this in our revised text (Lines 123 - 128; 198 - 199).
We are unable to comment on the biological significance of these changes, and we have refrained from speculating about the impact of these specific changes in the paper. Nevertheless, it is well known that subtle changes in cell wall architecture can have dramatic effects on a variety of phenotypes including cell wall elasticity, stress resistance and immune recognition [19,20,44].
Point 4:
The outer cell wall thickness increased in several alternative carbon sources, however in the case of the ethanol again the difference is about 12% and might not be biologically relevant.
Response 4:
Fair point. We now address this on lines 126 - 127 and 201.
Regarding biological significance, once again we were careful to discuss changes in the cell wall in general terms. We did not speculate about the impact of these specific changes. Please see our response to the previous point (Point 3).
Point 5:
The EM images to determine the cell wall thickness is a good experiment, but other assay could help to further prove this observation. Different assays (such as biochemical fractionation or staining of b-glucans/chitin and mannose) can be used to further confirm those results.
Response 5:
This is an interesting suggestion, which is greatly appreciated. This will be the potential focus of future studies.
Point 6:
The most interesting result is the resistance to amphotericin B in the alternative carbon sources. Repeat the experiment with another antifungal drug will be interesting to see whether this is a general resistance feature.
Response 6:
Thanks for the great suggestion, which is greatly appreciated. However, this idea, which lies beyond the scope of this study, would be interesting to study in the future. Based on observations in C. albicans [2], changes in carbon source are unlikely to affect resistance to other antifungal drugs in the same way.
Point 7:
Figure 5, the authors assay the resistance of C. glabrata to H2O2. Here the authors grow the cells for a few hours in the alternative carbon source and then plate them in YPD+ H2O2. This kind of experiment are frequently performed in a different way. The cells quickly adapt to the glucose in the YPD plates. Therefore, they should plate the cells in YP+ alternative carbon source+ H2O2 and compare to the YPD and YPD+ H2O2 plates. Nevertheless, this will imply to compare cells in different plates and still will not be ideal. Therefore, growing curves (as in Fig.1) in presence of alternative carbon sources and with or without H2O2 might be more appropriate in this case.
Response 7:
As described (Please refer to lines 316 - 323), for this experiment, C. glabrata cells were pre-grown on alternative carbon sources before being treated with H2O2 for only 3 hours. After 3 hours of incubation, the cells were washed and plated on YPD (there is no H2O2 in the YPD agar) to examine the survival of alternative carbon sources grown-C. glabrata cells following H2O2 treatment.
Point 8:
The authors do not perform pathogenic assays after growing cells in alternative carbon sources. In some species, infections using vegetables or fruits are commonly use in the laboratory. I am not sure how easy is to perform this assay with C. glabrata, but will be pretty interesting to perform.
Response 8:
The standard assay of pathogenesis for this human fungal pathogen is performed in mouse model. Indeed, we showed previously that a key enzyme (ICL1) required for alternative carbon utilisation is essential for the virulence of C. glabrata in the mouse model of invasive candidiasis (Chew et al. (2019) Sci Rep 2019, 9: 2843). Unfortunately, equivalent experiments were beyond the scope of this study. Hopefully, this will be possible in the future.

Round 2
Reviewer 2 Report
The authors only performed the minor revisions and avoid to do any of the suggested experiments. I have the same concerns that before since no real improvement (apart from the minor changes) have been introduced in the manuscript.
I did perfectly understand how experiment in Figure 5 (point 7) was done. But I do not consider that it was done properly. The grow on different carbon source including H2O2 in the media is the appropriate experiment to assess stress response since the cells quickly adapt after washing away the H2O2 and no effect is normally seen (as the authors saw, but because the experiment is not the most appropriate). This is an easy assay and can be done before publication.
Author Response
RESPONSES TO REVIEWER 2 COMMENTS (Revision 2)
Dear Editor,
Thank you for sending us the Reviewer’s helpful comments. We have taken them fully into account in our revised manuscript (changes to the text highlighted in yellow). We feel the changes have strengthened the paper, which we hope is now suitable for publication.
Our responses to the Reviewer’s comments are as follows:
Point 1:
The authors only performed the minor revisions and avoid to do any of the suggested experiments. I have the same concerns that before since no real improvement (apart from the minor changes) have been introduced in the manuscript.
I did perfectly understand how experiment in Figure 5 (point 7) was done. But I do not consider that it was done properly. The grow on different carbon source including H2O2 in the media is the appropriate experiment to assess stress response since the cells quickly adapt after washing away the H2O2 and no effect is normally seen (as the authors saw, but because the experiment is not the most appropriate). This is an easy assay and can be done before publication.
Response 1:
We have now performed new oxidative stress assays using the method suggested by the Reviewer. These have provided new insight into the effects of carbon source on oxidative stress resistance, and we have revised the manuscript based on these new results (changes and new additions are highlighted in yellow) (Lines 4, 31-33, 156-167, 220-228, 235, 324-330, 467-481).

Round 3
Reviewer 2 Report
The authors provided new data growing cells in presence of H2O2 as suggested (without washing it away). Indeed, a difference is observed when the stress is maintain as suggested. My main concerned is fixed and the manuscript is ready for publication.